# Adherence to the Mediterranean Diet and Chronic Disease in Australia: National Nutrition and Physical Activity Survey Analysis

**DOI:** 10.3390/nu12051251

**Published:** 2020-04-28

**Authors:** Yasmine S. Aridi, Jacqueline L. Walker, Eugeni Roura, Olivia R. L. Wright

**Affiliations:** 1School of Human Movement and Nutrition Sciences, The University of Queensland, Brisbane, QLD 4072, Australia; j.walker3@uq.edu.au (J.L.W.); o.wright@uq.edu.au (O.R.L.W.); 2Centre for Nutrition and Food Sciences, Queensland Alliance for Agriculture and Food Innovation, The University of Queensland, Brisbane, QLD 4072, Australia; e.roura@uq.edu.au

**Keywords:** nutrition, Mediterranean diet, chronic diseases, non-communicable diseases

## Abstract

The Mediterranean diet (MD) is linked to decreased risk of chronic disease, such as cardiovascular disease, obesity, hypertension, diabetes mellitus and cognitive disease. Given the health promoting aspects of this diet, we conducted a secondary analysis of data from the National Nutrition and Physical Activity Survey (NNPAS), which is the largest health study in Australia and the first nutrition-specific national-based study. The primary aim of this analysis was to determine the proportion of Australian adults adhering to the MD and to examine the association between adherence to the MD and markers of noncommunicable diseases, such as cardiovascular disease, diabetes mellitus and chronic kidney disease. Out of the 9435 participants included in the study (mean age = 48.6 ± 17.6 years), 65% were in the middle tertile of the MD score. Participants who were married, employed, of a high-socioeconomic level, nonsmokers, educated and had a healthy body mass index (BMI) and waist circumference were more likely to have higher adherence levels to the MD, which was associated with lower diastolic blood pressure (*p* < 0.05). Multivariate logistic regression analysis showed that, even after accounting for all possible confounders, higher adherence to the MD was associated with lower risk of dyslipidaemia, OR = 1.06 (1.01–1.10). In conclusion, this analysis is the first to assess adherence to the MD on a national level. Our results indicated that MD adherence may contribute to reducing the prevalence of dyslipidaemia, cerebrovascular disease and elevated blood pressure in a multi-ethnic, non-Mediterranean country.

## 1. Introduction

The Mediterranean diet (MD) is the traditional diet/lifestyle that people residing near the Mediterranean Sea follow. This area includes countries from Southern Europe, Northern Africa and the Middle East. Given that the MD is a traditional diet for many countries with literature demonstrating the various health benefits, in 2013, the United Nations Educational, Scientific and Cultural Organization added the MD to its “Representative List of the Intangible Cultural Heritage of Humanity” [1]. Specific consumption of MD foods differs among different Mediterranean countries; these patterns might also differ within the same country. These discrepancies are due to environmental factors, such as food availability and cultural traditions, however, the key components of this diet remain consistent across time and countries. The diet is characterized by a high intake of fruits, vegetables, fish, legumes, nuts and complex carbohydrates, as well as a low intake of poultry, red meats and sweets. The MD is a high-fat diet, with the majority of this fat coming from extra virgin olive oil, a healthy, monounsaturated fatty acid [2]. Cooking skills are passed on from one generation to the next, and eating is considered a social event that is enjoyed with family and friends [3]. The Mediterranean food pyramid was adapted from the Greek dietary pattern in the 1960s. Carbohydrate-rich foods (preferably whole grains), such as bread, pasta, cereal, bulgur, rice, potatoes and legumes, are at the base of this pyramid. Daily consumption of no less than five servings of fruits and vegetables, no more than two servings of dairy products and two servings of nuts, seeds, and olives is also recommended. Weekly, one should consume around two servings of poultry, no more than four eggs, at least two servings of fish/ seafood and legumes and no more than two servings of sweets. Red meats are at the very top of the pyramid, with consumption of no more than a few times per month [2].

The Mediterranean diet was shown to decrease the risk of developing noncommunicable diseases (NCDs), particularly diabetes mellitus, cardiovascular disease and kidney disease. An umbrella review of meta-analyses of cohort studies that examined the effect of the MD on chronic disease showed that participants with the highest levels of adherence to the MD had a 13–23% lower risk of developing diabetes and a 19–27% lower risk of developing cardiovascular disease (CVD) when compared to participants with the lowest adherence levels to the MD [4]. Despite these findings, the latest Cochrane review indicated only meagre evidence for the effectiveness of the MD on enhancing cardiovascular health due to low- to moderate-quality studies in primary prevention and poor evidence for secondary prevention [5]. The main challenge discussed in the Cochrane review involved limitations in data collection methods for dietary intake; this issue is further discussed in the limitations section in this paper. Diabetes mellitus occurs when the body is either not producing insulin at all, not producing enough insulin or is resistant to the insulin produced, contributing to increased blood and urine glucose levels as well as increased glucose stores in the liver. It is diagnosed by increased serum levels of fasting glucose or glycated haemoglobin levels [6]. Cardiovascular disease includes various diseases that affect the heart and blood vessels. Some of the most prevalent CVDs include stroke, coronary heart disease, heart failure, vascular disease and congenital heart disease. Some of the determinants of CVD include dyslipidaemia, hypertension, diabetes, tobacco use, low physical activity and increased body mass index [7]. Chronic kidney disease is characterized by abnormal kidney function or damage that lasts more than three months and is divided into five stages, namely, increased glomerular filtration rate, mild, moderate and severe chronic kidney disease (CKD) and end-stage renal disease (ESRD). The first two stages are diagnosed by the presence of albuminuria irrespective of glomerular filtration rate (GFR), whereas, the remaining three stages are diagnosed according to a glomerular filtration rate (GFR) of less than 60 mL/min. Some common risk factors for CKD include hypertension, increased BMI, age, diabetes, decreased physical activity and smoking [8].

The rates of NCDs are increasing worldwide. According to the Australian Institute of Health and Welfare, at least half of Australians are living with a chronic or noncommunicable disease [9]. The three major chronic lifestyle diseases, i.e., diabetes mellitus, cardiovascular diseases and chronic kidney disease, account for 25% of Australia’s national disease burden [10] and are all closely associated with one another. A recent report showed that the most common modifiable factors that increase the risk of developing these chronic diseases are increased body mass index, physical inactivity and poor dietary habits [11]. Given the prevalence of NCDs and the beneficial anti-inflammatory effects of the Mediterranean diet, the aim of the current study was to (i) determine the proportion of Australian adults adhering to the MD and (ii) examine the association between adherence to the MD for Australian adults and NCDs, such as cardiovascular disease, diabetes mellitus and CKD. This is the first study to assess adherence to the MD among Australians on a national level and to investigate its linkage to biomarkers of chronic disease.

## 2. Materials and Methods

### 2.1. Study Design and Sample Collection

This study was a secondary analysis of data collected as part of the National Nutrition and Physical Activity Survey (NNPAS). The NNPAS was conducted by the Australian Bureau of Statistics (ABS) between the years 2011 and 2013. Participants were approached by a primary approach letter sent via mail. The letter contained a briefing about the study as well as how information would be confidential in accordance with the Census and Statistics Act 1905. Prior to data collection, all planned questionnaires were field tested to ensure data quality. Additionally, participants willing to take part in the study completed informed consent forms and kept a copy. Briefly, data collection took place nationwide using systematic random sampling. The country was divided into areas, areas into states and states into strata. Each strata was divided into collection districts and each district into private residences. Detailed descriptions of these methods can be found elsewhere [12]. Data collection took place by trained and experienced interviewers and was done twice, the first time in a face-to-face interview and the second via telephone. A total of 9519 households completed all stages of data collection, however, only one adult (aged 18+ years) per household was included in the study [13].

All data collected during the face-to-face interviews and telephone calls were entered into the Australian Bureau of Statistics (ABS) basic Confidentialised Unit Record Files (CURF). The CURF contained all details related to the study population’s characteristics, dietary habits, biochemical tests and history of diseases. Given that the secondary analysis was done by members of The University of Queensland, the authors had full access to this dataset. The CURF data sets were exported and merged into the Statistical Package for Social Sciences (SPSS) (version 25.0.0, 2017, IBM, Chicago, USA) for further analysis. The data was then cleaned and checked for the presence of any outliers. Demographics of the sample population that were extracted from the CURF included age, sex, registered and social marital status, country of birth and year of arrival in Australia, Indigenous status, language mainly spoken at home, proficiency in English and country of birth of parents, highest education level attained, labour force status, income, housing and geographical characteristics (Table 1).

This analysis protocol was submitted to The University of Queensland’s Human Research Ethics Committee and deemed exempt from ethics review under the National Statement on Ethical Conduct in Human Research, reference number 2017001679.

### 2.2. Biochemical Data

Biochemical data recorded in the CURF were collected via blood or urine tests by trained and licensed phlebotomists. Participation in the biochemical tests was voluntary for all participants. The tests were performed at Sonic Healthcare collection clinics or via a home visit. Diabetes mellitus was assessed using serum levels of fasting plasma glucose and glycated haemoglobin (HbA1c). A person was considered to be diabetic if they had a fasting glucose level greater than or equal to 7.0 mmol/L or an HbA1c level greater than or equal to 6.5%. Cardiovascular disease status was assessed using serum levels of total cholesterol, high density lipoprotein (HDL) cholesterol, low density lipoprotein (LDL) cholesterol and triglycerides. HDL less than 1 mmol/L, LDL greater than or equal to 3.5 mmol/L, triglycerides greater than or equal to 2 mmol/L and total cholesterol greater than or equal to 5.5 mmol/L were defined as abnormal levels. A participant was considered to have dyslipidaemia if they were taking cholesterol-lowering medication, had a total cholesterol of ≥5.5 mmol/L, an HDL cholesterol of <1.0 mmol/L for men and <1.3 mmol/L for women, an LDL cholesterol of ≥3.5 mmol/L or triglycerides of ≥2.0 mmol/L. Systolic and diastolic blood pressure measurements were recorded during the interviews following recommended protocol and procedure. Hypertension was determined as having a blood pressure greater than or equal to 140/90 mmHg. The basal metabolic rate (BMR) was predicted according to the participant’s age, sex and weight. Lastly, a participant was deemed to have CKD if they had a GFR of less than 60 mL/min/1.73 m^2^ or if they had albuminuria levels of greater than 30 mg/g. Detailed classifications of these diseases can be found elsewhere [12].

### 2.3. Dietary Data

Dietary data were collected based on two interviewer-administered 24 h recalls, as well as additional questions related to usual dietary habits, such as food avoidance due to allergies, usual consumption of salt and water and supplement use. A detailed explanation of the dietary data collection process was previously published [14]. To ensure data quality, the NNPAS team used the Automated Multiple-Pass Method (AMPM) and provided participants with a food model booklet. The AMPM is an interviewer-based method to collect dietary data using 24 h recalls made up of five phases, namely, quick list, forgotten foods, time and occasion, detailed cycle and a final probe. It collects detailed data about the foods consumed, such as the name, weight, cooking style, time consumed and any additions to the food. This method was developed by the United States Department of Agriculture and is considered the gold standard for 24 h recall collection and was shown to minimize the effect of under-reporting [15,16]. Using this method, 24 h recalls were collected during the interview as well as 9 days later via telephone. One 24 h recall was collected for all participants in the NNPAS via telephone interview. An additional 24 h recall was conducted by telephone with 7585 participants who responded to the invitation [17]. Foods from the 24 h recalls were added into an ABS database and converted into specific recipes. A food model booklet was used to help participants visualize portion sizes. If participants did not complete the second 24 h recall, their first 24 h recall results were carried over in the analysis to minimize sample size loss.

Data from the 24 h recalls were used to calculate a Mediterranean diet adherence score from 0 to 9 in this work [18]. When researching the literature, Trichopoulou’s 0–9 MD score was one of the most widely used [19], which was the first MD score ever established to assess adherence to the traditional Greek Mediterranean diet. Briefly, each of the nine food groups were given a score of either 0 or 1. Favourable food (fruits and nuts, vegetables, cereals, legumes), intake equal to or higher than the sample population’s sex-specific median was given a 1, while intake below the median was given a 0. Unfavourable foods (meats/meat products, poultry and dairy products) with intake less than the sample population’s sex-specific median was given a 1, while intake higher than the median was given a 0. Ethanol intake of 10–50 g/day for men and 5–25 g/day for women was given a score of 1, as these values represent the recommended guidelines of alcohol consumption according to Trichopoulou’s criteria [20]. The type of alcohol was not considered. If the ratio of monounsaturated fatty acid to saturated fatty acid was greater than 1, it was also given a score of 1. In total, Trichopoulou’s Mediterranean dietary score ranged from 0 (lowest adherence) to 9 (highest adherence) [20]. Given that the Mediterranean dietary score is not a scalar measurement, scores were divided into tertiles.

### 2.4. Statistical Analysis

The Statistical Package for Social Sciences (SPSS, version 25.0.0, 2017, IBM) was used for data cleaning, management and analyses. All results were stratified according to MD score tertiles. Continuous data were reported as the median, mean and standard deviations (Sd). Normally distributed variables, such as BMI, waist circumference, diastolic blood pressure (BP), systolic BP, total energy, percentage of energy from protein, percentage of energy from fat, percentage of energy from carbohydrates and percentage of energy from fiber, were reported as the mean and Sd. Comparisons between MD score tertiles and continuous variables were completed using the one-way analysis of variance (ANOVA) test. Categorical variables were reported as numbers and percentages. Associations between these variables and the MD score tertiles were established using the Chi-square test. To identify predictors of disease, multivariate logistic regression was performed considering the variables found to be statistically significant at the bivariate level. The Chi-square test was used to determine the odds ratios (ORs) with corresponding 95% confidence intervals (CI). A *p*-value of <0.05 indicated statistical significance. Covariate analyses were adjusted according to age (years), gender (reference: male), marital status (reference: married), country of birth (reference: Australia), Socio-Economic Indexes for Areas (SEIFA) (reference: lowest 20%), labour force status (reference: employed), smoking (reference: no) and education level (reference: school). Covariates were determined according to the demographics of the sample stratified by the MD score.

## 3. Results

Table 1 shows the demographics of the sample population. A total of 9435 participants were included in this analysis. Participants had a mean age of 48.6 ± 17.6 years. Marital status had a significant impact on MD scores, with greater adherence to the MD indicated by married participants (*p*-value < 0.05). More precisely, when examining the highest MD score tertile compared to the lowest tertile, married participants accounted for 57.9% of the participants, whereas nonmarried participants made up 42.1%. Likewise, employed participants and participants within the highest 20% of SEIFA accounted for 68.4% and 29.8% of the highest MD score tertile, respectively (*p* < 0.0001). When analysing participants’ anthropometrics, those with low MD adherence scores were more likely to be overweight or obese by BMI classification (*p* < 0.0001). Additionally, participants within the highest MD score tertile had lower waist circumference values for both women and men compared to the lowest tertile (*p* < 0.01 and *p* < 0.0001, respectively). Analysis of participants’ general dietary compositions showed that those within the highest MD score tertile exhibited lower percentages of energy from protein and fat and higher percentages from carbohydrates, as well as higher total energy intakes (*p* < 0.0001). No significant differences (*p* > 0.05) were observed across the MD adherence tertiles related to gender or age. The consumption of legumes and fish was very low in our sample; almost 50% of the sample population did not consume any fish or legumes. 

Prevalence of diabetes mellitus was assessed using fasting plasma glucose and HbA1c percentage values. The MD score was not associated with any diabetes mellitus indicators (Table 2). Table 2 shows the association between MD score and chronic kidney disease (CKD). The results indicated that 451 participants had CKD, however, rates of CKD did not statistically differ between MD score tertiles.

Table 3 shows the association between MD score and lipid profile markers. Of the overall sample population, 36.9% exhibited abnormal total cholesterol status, 36.1% showed elevated LDL levels and 69.2% suffered from dyslipidaemia. Our results showed that adherence to the MD was associated with lower rates of abnormal total cholesterol status (*p* < 0.05). Similarly, increased adherence to the MD was associated with improved LDL status (*p* < 0.05) and decreased rates of overall dyslipidaemia (*p* < 0.05).

The association between the MD score and heart disease is shown in Table 4. The cohort of participants included 185 individuals with cerebrovascular disease. While 24.3% belonged to the lowest MD adherence tertile, only 7.5% belonged to the highest MD adherence group. The MD was shown to be significantly associated with rates of cerebrovascular disease (*p* < 0.05). The majority of the sample, 40.4%, had normal blood pressure, 36.1% were pre-hypertensive and 23.3% were hypertensive. The results approached statistical significance for the association between MD adherence score and blood pressure (*p* = 0.07). A similar association was also observed between the MD tertiles and systolic blood pressure (*p* = 0.07). Diastolic blood pressure was significantly associated with MD tertiles, with the highest MD adherence levels linked to lower blood pressure (*p* < 0.05).

All possible confounders identified in Table 1 were added into the multivariate analysis in order to examine the association between MD score tertiles and CVD, CKD, dyslipidaemia and diabetes mellitus. Possible confounders included age, sex, marital status, country of birth, education, labour force status, SEIFA and smoking. These results are shown in Table 5. Bivariate analysis showed that the MD was significantly associated with total cholesterol, fasting LDL cholesterol and dyslipidaemia. The association with LDL cholesterol and dyslipidaemia disappeared after including other confounders, such as SEIFA, smoking, education, marital status and age. However, adherence to the MD remained associated with total cholesterol (*p* < 0.01). Similar to the bivariate analysis, adherence to the MD was not associated with increased risk of diabetes mellitus or CKD in the multivariate analysis.

## 4. Discussion

Our most striking finding was the significant effect of adhering to the MD on cardiovascular health parameters. In our sample population of Australian adults, bivariate Chi-square analyses showed that higher adherence to the MD was associated with (i) lower total cholesterol levels (*p* < 0.05), (ii) lower LDL cholesterol levels (*p* < 0.05) and (iii) a lower risk of dyslipidaemia (*p* < 0.05). Likewise, higher adherence to the MD was associated with lower risk of developing cerebrovascular disease and lower systolic blood pressure. A borderline statistically significant association was observed between higher adherence to the MD and diastolic blood pressure (*p* = 0.04). Adherence to the MD was also associated with having a healthier BMI (*p* < 0.001) and waist circumference (*p* < 0.001), regardless of gender. Our analysis also determined factors that may affect adherence to the MD among the Australian population, such as age, gender, marital status, country of birth, SEIFA, labour force status, smoking status and educational level. Finally, multivariate logistic regression showed that, even after accounting for all possible confounders, an increase of one unit in the MD adherence score was associated with a lower risk of dyslipidaemia (OR 1.06, 95% CI 1.01–1.10).

Our results were consistent with current evidence regarding the MD and its cardiovascular effects. Grosso et al., in their meta-analysis of 11 studies, showed that participants with the highest MD adherence scores exhibited significantly lower risk of developing CVD and CHD [21]. A recent meta-analysis of six trials showed that the MD had a significant positive effect on CVD parameters [22]. Another study showed that after 10 years of adherence to the MD, participants with the highest MD scores were 26% less likely to suffer from myocardial infarction and 22% less likely to suffer from a stroke [23]. Our findings demonstrating that higher adherence to the MD was significantly associated with lower total cholesterol levels even after accounting for all possible confounders was consistent with a meta-analysis that showed that MD was associated with decreased concentrations of total cholesterol and triglycerides and increased concentrations of high-density lipoprotein [24]. Additionally, our results were consistent with another study where the MD was associated with lower levels of systolic and diastolic blood pressure [24]. A recent study showed that participants with the highest levels of adherence to the MD compared to those who were the least adherent had significantly lower systolic (133.3 ± 23.7 vs. 135.3 ± 14.9 mmHg) and diastolic (78.6 ± 8.5 vs. 80.7 ± 8.7 mmHg) BP values [25]. Likewise, a meta-analysis that included more than 7000 participants also showed that people abiding by the MD guidelines for one year exhibited significantly lower systolic and diastolic blood pressure values [26].

A surprising result from our study was the lack of association between blood glucose parameters or diabetes prevalence and adherence to the MD. This may be because only a small percentage of the study sample (6.6%) had diabetes mellitus, which was comparably low to other studies. Additionally, insulin resistance was not measured in the NNPAS survey. In a meta-analysis of nine studies, adherence to the MD was shown to significantly reduce levels of glycated haemoglobin (HbA1c), fasting glucose and insulin among patients with type 2 diabetes [24]. In another meta-analysis including only cohort studies and randomised controlled trials (RCTs), Schwingshackl et al. [27] showed that participants adhering more to the MD had 19% less risk of developing diabetes mellitus type 2 later in life, (95%CI 0.73–0.90). These results were further strengthened by an RCT conducted in Spain, which showed that participants who adhered to the MD with extra virgin olive oil had significantly lower risk (51%) of developing DM compared to those who did not adhere to the diet [28].

Our results showed that higher adherence to the MD was associated in a statistically significant manner with lower BMI (*p* < 0.01) and waist circumference (*p* < 0.01). Other studies showed statistically significantly lower waist circumference values in participants adhering to the MD [29,30,31]. Results from a prospective cohort study in 10 European countries showed that participants with the highest MD adherence scores were 10% less likely to be overweight or obese compared to participants with the lowest scores [32]. In a meta-analysis of six trials, including overweight and obese participants, Nordmann et al. showed that the MD induced more weight loss and greater decrease in BMI compared to a low-fat diet [22]. Results were similar in a more recent meta-analysis, where the MD induced significant weight loss in obese participants [33].

No significant association was observed among MD adherence and CKD. Various tests exist to measure CKD, including GFR and albumin/creatinine ratio; hence, the use of either test could affect the association between MD and CKD. Moreover, the association between CKD and the MD has not yet been extensively studied; more studies are required in this field. Although components of the MD, such as omega 3, fruits, vegetables and whole grains, are well-known to protect against CKD, more prospective studies are required to examine the MD as a whole [34]. However, a recent cross-sectional German study used a food frequency questionnaire (FFQ) in CKD patients and found that the MD was associated with better GFR levels even after accounting for all possible confounders [35]. Another six-year follow-up study found that participants with higher rates of MD adherence showed around a 50% lower risk of developing CKD [36]. Almost half of our sample did not consume any fish or legumes, which might have affected our results given that both were shown to be associated with a lower risk of CKD [37], diabetes mellitus [38,39,40], cardiovascular disease [41,42] and dyslipidaemia [37,43,44,45]. If our sample consumed more fish and legumes, we could have detected an even stronger association between MD and these diseases.

Some proposed mechanisms by which the MD affects the risk of developing noncommunicable diseases are its anti-inflammatory and antioxidative aspects. The anti-inflammatory effect seems to be the mediator between MD and diabetes mellitus by improving insulin sensitivity via better endothelial function by increasing flow-mediated dilation and decreasing intercellular adhesion [46]. Another proposed mechanism by which the MD is associated with lower risk of diabetes mellitus is through decreasing the rates of obesity [47], as excessive body fat is linked to insulin resistance [47]. Another potential mechanism by which that the MD protects against diabetes mellitus is through its high monounsaturated fatty acid content, fiber and antioxidant-rich foods, including fruits, vegetables, extra virgin olive oil, nuts and seeds [48,49]. This anti-inflammatory effect extends to protect against kidney dysfunction [50], with the association between MD and chronic kidney disease attributed to the presence of foods such as nuts, legumes and fish [51]. Moreover, the high fiber content of the MD was shown to improve GFR levels by decreasing nephron workload [50]. Lastly, the antioxidative factors in the MD protect kidney function by improving endothelial function and protecting against major risk factors for CKD, such as obesity and diabetes mellitus [50]. These anti-inflammatory and antioxidative aspects of the MD were also shown to decrease the risk of cardiovascular disease. Polyphenols found in this diet, specifically in extra virgin olive oil and red wine, were directly linked to better cardiovascular health by improving serum lipid profiles and blood pressure [52]. Additionally, the extensive use of extra-virgin olive oil in the MD and the limited consumption of saturated fatty acids protects against CVD [53]. Also, the MD protects against high blood pressure, a major risk factor for CVD, given that it exerts a vasoprotective effect by improving endothelial cell nitric oxide production [54].

### Strengths and Limitations

This was the first study to assess adherence to the MD among the Australian population on a national level, comprising a large dataset collected in a robust, thorough and scientific manner. Additionally, given that participants in the NNPAS represent Australians on a national level, results from this study can be extrapolated to the overall Australian population. Another strength of this study is that it was based on measured indicators of chronic diseases, and all blood/urine tests were conducted according to standardized protocols by trained/licensed professionals.

A limitation of this study is that dietary data was based on one or two 24 h recalls. Dietary intake varies from one day to another, thus regular dietary intake is hard to predict. The two recalls may not have represented usual dietary intake for the study sample. Furthermore, if participants did not complete the second 24 h recall, their first 24 h recall results were carried over to the analysis to minimize sample size loss. We wanted to be able to include the full sample in our analysis and not create any biases by excluding people who did not complete the second dietary recall. Despite this, dietary data was collected using the automated multiple pass method for 24 h recalls. This method is the gold standard for data collection given that it minimizes interviewer bias and reporting bias by standardizing data collection and providing participants with visual aids for portion sizes [54]. The study was a cross-sectional design which prevented us from detecting a cause/effect relationship between adherence to the MD and the prevalence of chronic diseases and biomarkers. Under-reporting was calculated by calculating the ratio of energy intake (EI) to basal metabolic rate (BMR). A cut-off value of 0.9 was used based on Goldberg et al.’s method [55]. The highest percentage (27 percent) of under-reporting was among the participants in the lowest tertile of the MD score (Table 1). This group of under-reporters exhibited the highest BMI and waist circumference values, which was consistent with previous reports showing that overweight and obese individuals were the most likely to under-report their dietary intake [56]. Lastly, data for the NNPAS were collected between 2011 and 2012, hence dietary intake among Australians may have changed since this data collection. However, the NNPAS remains the most recent survey that represents dietary consumption among Australians.

## 5. Conclusions

Our secondary analysis showed that that higher adherence to the MD was associated with lower total cholesterol, lower low-density lipoprotein, lower diastolic blood pressure and lower levels of dyslipidaemia in a large sample of Australians. Participants who were married, employed, of a high socioeconomic level, nonsmokers, educated or who had a healthy body weight and waist circumference were more likely to have higher adherence levels to the MD. These findings add to the current body of literature as they demonstrated the potential health benefits for Australians in following a Mediterranean dietary pattern, but also highlighted groups of the population who require further support and assistance to adhere to this lifestyle choice. Future consultation with and nutritional intervention in these at-risk groups to help them adhere to the MD pattern may be beneficial in preventing the development of chronic disease and improving the health of the general population.

Given the ethnic diversity of the people residing in Australia, many of whom are of Mediterranean origin, it would be beneficial to assess adherence to the Mediterranean diet in association with factors that affect this adherence in Australia. Additionally, stratifying the data in Australian research according to the ethnicity of participants would provide a more comprehensive understanding of the applicability of the Mediterranean diet in a multicultural setting. Future research involving prospective longitudinal studies and large group intervention studies would assist the advancement of this field. Lastly, given the well-known health benefits of the MD, future studies should examine factors that affect adherence to this dietary pattern in a multi-ethnic community to identify adherence challenges and ways to overcome them.

## Figures and Tables

**Table 1 nutrients-12-01251-t001:** Demographics of the sample population in the National Nutrition and Physical Activity Survey (NNPAS) study.

			Mediterranean Diet Score			
Total Study Sample, *n* (%)		Overall Sample	Tertile 1 (0–3) *N* (%) = 1930 (20.5)	Tertile 2 (4–6) *N* (%) = 6190 (65.6)	Tertile 3 (7–9) *N* (%) = 1315 (13.9)	*p*-value
**Gender**	Female	5106 (54.1)	1067 (55.3)	3353 (54.2)	686 (52.2)	0.21
Age (years)	Mean (±SD)	48.6 ± 17.6	48.8 ± 18.16	48.5 ± 17.6	48.8 ± 16.5	0.74
	18–39	3289 (34.9)	663 (34.4)	2190 (35.4)	436 (33.2)	0.11
	40–64	4043 (42.9)	830 (43.0)	2607 (42.1)	606 (46.1)	
	65+	2103 (22.3)	437 (22.6)	1393 (22.5)	273 (20.8)	
Social marital status	Married	4958 (52.5)	936 (48.5)	3260 (52.7)	762 (57.9)	**<0.0001**
BMI (kg/m^2^)	Mean (±SD)	27.5 ± 5.5	28.1 ± 5.9	27.5 ± 5.5	26.4 ± 4.8	**<0.0001**
	Underweight	121 (1.3)	27 (1.7)	79 (1.5)	15 (1.3)	**<0.0001**
	Healthy weight range	2736 (29.0)	475 (30.2)	1788 (34.2)	473 (40.8)	
	Overweight	2898 (30.7)	551 (35.0)	1892 (36.2)	455 (39.3)	
	Obese	2203 (23.3)	522 (33.1)	1465 (28.0)	216 (18.6)	
Country of birth	Australia	6714 (71.2)	1540 (79.8)	4432 (71.6)	742 (56.4)	**<0.0001**
	Main English-Speaking Countries	1155 (12.2)	226 (11.7)	721 (11.6)	208 (15.8)	
	Others	1566 (16.6)	164 (8.5)	1037 (16.8)	365 (27.8)	
Arrival to Australia	Born in Australia	6714 (71.2)	1540 (82.6)	4432 (75.7)	742 (62.2)	**<0.0001**
	Arrived 1985 or before	1323 (14.0)	224 (12.0)	870 (14.9)	229 (19.2)	
	Arrived 1986–1990	266 (2.8)	34 (1.8)	161 (2.7)	71 (6.0)	
	Arrived 1991–1995	171 (1.8)	17 (0.9)	108 (1.8)	46 (3.9)	
	Arrived 1996–2000	195 (2.1)	24 (1.3)	127 (2.2)	44 (3.7)	
	Arrived 2001–2005	246 (2.6)	26 (1.4)	160 (2.7)	60 (5.0)	
Labour force status	Employed	6009 (63.7)	1134 (58.8)	3975 (64.2)	900 (68.4)	**<0.0001**
SEIFA	First quintile	1778 (18.8)	434 (22.5)	1152 (18.6)	192 (14.6)	**<0.0001**
	Second quintile	1961 (20.8)	481 (24.9)	1245 (20.1)	235 (17.9)	
	Third quintile	1873 (19.9)	369 (19.1)	1268 (20.5)	236 (17.9)	
	Fourth quintile	1666 (17.7)	284 (14.7)	1122 (18.1)	260 (19.8)	
	Highest quintile	2157 (22.9)	362 (18.8)	1403 (22.7)	392 (29.8)	
Waist circumference (cm)	Mean (±SD)	93.1 ± 14.5	94.5 ± 14.7	93.1 ± 14.6	90.7 ± 13.3	**<0.0001**
Waist circumference (cm)—female	Increased risk (more than or equal to 80 cm)	2891 (30.6)	618 (74.0)	1896 (69.2)	377 (66.1)	**0.004**
Waist circumference (cm)—male	Increased risk (more than or equal to 94 cm)	2333 (24.7)	491 (66.9)	1537 (62.1)	305 (52.3)	**<0.0001**
Currently smokes	Yes	1785 (18.9)	509 (26.4)	1130 (18.3)	146 (11.1)	**<0.0001**
Education level	School	7050 (75.2) 9372	1634 (84.7)	4602 (74.3)	814 (61.9)	**<0.0001**
	Bachelor degree	1615 (17.2)	205 (10.6)	1083 (17.5)	327 (24.9)	
	Postgraduate degree	770 (8.2)	91 (4.7)	505 (8.2)	174 (13.2)	
Percentage of energy from protein (%)	Mean (±SD)	18.6 ± 5.5	19.2 ± 6.09	18.6 ± 5.5	17.7 ± 4.4	**<0.0001**
Percentage of energy from total fat (%)	Mean (±SD)	30.6 ± 7.7	31.7 ± 8.0	30.5 ± 7.6	29.5 ± 7.2	**<0.0001**
Percentage of energy from carbohydrate (%)	Mean (±SD)	43.2 ± 9.8	42.7 ± 10.4	43.2 ± 9.8	44.2 ± 8.9	**<0.0001**
Percentage of energy from alcohol (%)	Mean (±SD)	12.8 ± 9.9	15.5 ± 13.3	12.8 ± 9.6	10.6 ± 7.1	**<0.0001**
Energy intake (Kj)	Mean (±SD)	8319.3 ± 3204.4	7703.1 ± 3167.9	8348.4 ± 3199.3	9085.0 ± 3103.7	**<0.0001**
Rates of under-reporting	EI: BMR < 0.9	1575 (19.7)	429 (27.0)	1014 (19.3)	132 (11.4)	**<0.0001**

BMI: body mass index; SEIFA: Socio-Economic Indexes for Areas; SD: standard deviation; EI: energy intake; BMR: basal metabolic rate. Significant *p*-values (<0.05) are shown in bold.

**Table 2 nutrients-12-01251-t002:** Association between Mediterranean diet score, diabetes mellitus prevalence and chronic kidney disease prevalence.

			Mediterranean Diet Score			
	Prevalence of Diabetes Mellitus	All	Tertile 1 (0–3) (*n* (%))	Tertile 2 (4–6) (*n* (%))	Tertile 3 (6–9) (*n* (%))	*p*-value
Fasting plasma glucose (mmol/L)	≥7.0 mmol/L	211 (6.6)	36 (6.1)	146 (7.0)	29 (5.7)	0.49
HbA1c (%)	≥6.5%	253 (6.7)	44 (6.3)	176 (7.1)	33 (5.6)	0.39
	**Prevalence of CKD**					
CKD		451 (12.8)	89 (14.1)	300 (12.8)	62 (11.2)	0.34

HbA1c: glycated haemoglobin; CKD: chronic kidney disease.

**Table 3 nutrients-12-01251-t003:** Association between the Mediterranean diet score and plasma lipid profiles (cholesterol, HDL, triglycerides and LDL).

			Mediterranean Diet Score			
		All	Tertile 1 (0–3) (*n* (%))	Tertile 2 (4–6) (*n* (%))	Tertile 3 (6–9) (*n* (%))	*p*-value
Total cholesterol (mmol/L)	Normal (<5.5 mmol/L)	2377 (63.1)	411 (59.3)	1577 (63.5)	389 (65.7)	**0.04**
HDL cholesterol (mmol/L)	Normal (≥1.0 mmol/L)	3353 (89.0)	615 (88.7)	2206 (88.8)	532 (89.9)	0.75
Fasting triglycerides (mmol/L)	Normal (<2.0 mmol/L)	2681 (84.4)	488 (83.1)	1757 (84.3)	436 (85.8)	0.47
Fasting LDL cholesterol (mmol/L)	Normal (<3.5 mmol/L)	2006 (63.9)	350 (60.3)	1346 (65.5)	310 (61.4)	**0.03**
Dyslipidaemia	Does not have dyslipidaemia	965 (30.8)	152 (26.3)	648 (31.6)	165 (32.8)	**0.03**

HDL: high-density lipoprotein; LDL: low-density lipoprotein. Significant *p*-values (<0.05) are shown in bold.

**Table 4 nutrients-12-01251-t004:** Association between Mediterranean diet score and heart disease (heart failure, cerebrovascular disease hypertension).

			Mediterranean Diet Score			
		All	Tertile 1 (0–3) (*n* (%))	Tertile 2 (4–6) (*n* (%))	Tertile 3 (6–9) (*n* (%))	*p*-value
Heart failure or other heart disease	Yes	93 (1.0)	17 (0.9)	67 (1.1)	9 (0.7)	0.36
Cerebrovascular disease	Yes	185 (2.0)	45 (2.3)	126 (2.0)	14 (1.1)	**0.02**
Blood Pressure	Normal (<120/80 mmHg)	3282 (40.4)	642 (39.9)	2135 (40.2)	505 (43.3)	**0.07**
	Pre-hypertensive (120/80–140/90)	2930 (36.1)	575 (35.7)	1928 (36.3)	427 (36.6)	
	Hypertensive (≥140/90 mmHg)	1895 (23.3)	393 (24.4)	1249 (23.5)	235 (20.1)	
Diastolic blood pressure (mmHg)		76.8 ± 11.1	77.1 ± 10.9	76.9± 11.3	76.1 ± 10.6	**0.04**
Systolic blood pressure (mmHg)		123.3 ± 19.1	123.6 ± 18.8	123.5 ± 19.3	122.1 ± 18.4	**0.07**

Significant *p*-values (<0.05) are shown in bold.

**Table 5 nutrients-12-01251-t005:** Multivariate analysis of potential predictors of diseases.

Variables	OR (95% CI)	*p*-value
Diabetes mellitus (reference: no)		
MDS Score	1.02 (0.93–1.12)	0.67
CKD (reference: no)		
MDS Score	0.99 (0.9–1.06)	0.82
Heart failure or other heart disease (reference: no)		
MDS Score	0.98 (0.85–1.12)	0.79
Dyslipidaemia (reference: no)		
MDS Score	0.95 (0.90–1.00)	0.09
Total cholesterol mmol/L) (reference: abnormal)		
MDS Score	1.06 (1.01–1.10)	0.009
HDL cholesterol mmol/L) (reference: abnormal)		
MDS Score	1.02 (0.95–1.09)	0.51
Fasting triglycerides (mmol/L) (reference: abnormal)		
MDS Score	1.02 (0.96–1.08)	0.48
Fasting LDL cholesterol (mmol/L) (reference: abnormal)		
MDS Score	1.02 (0.98–1.07)	0.23

MDS score (increase by 1 unit). Confounders included in this analysis were age, sex (reference: male), marital status (reference: married), country of birth (reference: Australia), SEIFA (reference: lowest 20%), Labour force status (reference: employed), smoking status (reference: no), and education level (reference: school). OR: odds ratio. MDS: Mediterranean dietary score. Significant *p*-values (<0.05) are shown in bold.

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
