# Peer review of "Adherence to the Mediterranean Diet and Chronic Disease in Australia: National Nutrition and Physical Activity Survey Analysis"

_nutrients, 2020, doi:10.3390/nu12051251_

Round 1

Reviewer 1 Report

The authors present an important study in the Mediterranean diet and the risk of various chronic diseases and their biomarkers. The advantage of the study is their national level and large sample size.

Unfortunately, the authors did not avoid numerous shortcomings in the preparation of the manuscript. There are important study limitations. The authors did not provide information about the way of data collection - we can be supposed that were only declared. There a lack of information that the analyzed diseases were diagnosed and for how long before the participation in the study, whether the participants took chronic medications, dietary supplements, etc. - all these variables could impact on the dietary and biochemical data and the final results. Besides, the authors used a 24-hour recall to assess food intake - which may not have reflected the participants' usual diet. The manuscript needs to be completed with necessary methodological details. The manuscript requires English correction using professional editing and proof-reading service. Details comments were given in the attached file.

Author Response

We would first like to thank the reviewer for taking the time to review this paper. We appreciate your comments/suggestions and we we did our best to address them all. All modifications were done using track changes on the manuscript. Please find attached a doc. that addresses each of your comments

Many thanks

Reviewer 2 Report

The study is a secondary analysis of data from the National Nutrition and Physical Activity Survey, that evaluates the association with adherence to the Mediterranean diet and markers of chronic non-communicable diseases in the Australian adult population.

The strong point of the article is the vast number of subjects enrolled (9435 participants).
Limits are as follows:
1) The data were collected from 2011 to 2013, approximately 10 years
before their publication.
Therefore, they provide a non-current photograph of the situation but relative to about 10 years earlier.
2) In the study design there are a number of critical issues:
to monitor chronic kidney disease (CKD), reference is made exclusively to
GFR <60 ml / min/1.73m2, without taking into consideration the presence of albuminuria.
Therefore,
the first two stages of CKD which are characterized by a GFR> 60 ml/min/1.73m2
but by the presence of albuminuria are excluded a priori.

Furthermore, the authors generally refer to cardiovascular diseases, examining
only the monitoring of systolic and diastolic blood pressure.

It is not clear when blood pressure values were monitored and how they were monitored. Did the
subjects examined take antihypertensive therapy or not?
The
authors make no reference to antihypertensive therapy.

To evaluate the possible cardiovascular disease, the authors could have calculated the atherogenic indices.
Minor comments.
About the Mediterranean Diet, the authors refer to olive oil while they
should have indicated extra virgin olive oil.

Correct in the text “diabetes” with “diabetes mellitus”.

Author Response

We would first like to thank the reviewer for taking the time to review this paper. We appreciate your comments/suggestions and we we did our best to address them all. All modifications were done using track changes on the manuscript. Please find attached a doc. that addresses each of your comments

Many thanks

Reviewer number 2

The study is a secondary analysis of data from the National Nutrition and Physical Activity Survey, that evaluates the association with adherence to the Mediterranean diet and markers of chronic non-communicable diseases in the Australian adult population.

The strong point of the article is the vast number of subjects enrolled (9435 participants).

Limits are as follows:

1) The data were collected from 2011 to 2013, approximately 10 years

before their publication. Therefore, they provide a non-current photograph of the situation but relative to about 10 years earlier.

This might be true; however, this is the most current data in Australia on a national level. Analyzing data that has been collected 7 years ago is very common in the literature.

2) In the study design there are a number of critical issues:

to monitor chronic kidney disease (CKD), reference is made exclusively to

GFR <60 ml / min/1.73m2, without taking into consideration the presence of albuminuria. Therefore,

the first two stages of CKD which are characterized by a GFR> 60 ml/min/1.73m2

but by the presence of albuminuria are excluded a priori.

Thank you for noting that. Having stages 1 and 2 kidney diseases is not considered chronic unless you have albuminuria. We took this into account when calculating those who had stages 1 and 2; as for stages 3 onwards, we used GFR<60 ml / min/1.73m2. We understand that this has not been well noted on the table, so we have now updated the table and the text.

Furthermore, the authors generally refer to cardiovascular diseases, examining

only the monitoring of systolic and diastolic blood pressure.

It is not clear when blood pressure values were monitored and how they were monitored. Did the

subjects examined take antihypertensive therapy or not? The

authors make no reference to antihypertensive therapy.

This has been added to the Biochemical data section. Blood pressure was recorded during the interview following standardized procedure, and two measurements were taken and then averaged. The use of medications was not added in the analysis.

To evaluate the possible cardiovascular disease, the authors could have calculated the atherogenic indices.

Minor comments.

About the Mediterranean Diet, the authors refer to olive oil while they should have indicated extra virgin olive oil.

Done

Correct in the text “diabetes” with “diabetes mellitus”.

Done

Round 2

Reviewer 1 Report

Dear Authors,

the manuscript has been significantly improved; however, it is required revision. Unfortunately, not all comments have been considered with accuracy.

Details comments were given in the attached file.

Author Response

We would like to thank the reviewer for taking the time to review this paper. We hope we have succeeded in replying to all your comments. Kindly find attached a word doc. that replies to your feedback

Thank you for all of the comments on our paper. Please find a table below outlining all of our most recent changes, justification and line numbers for where changes have been made in the document.

Reviewer’s Comments

The manuscript has been significantly improved; however, it is required minor revision before publication. Unfortunately, not all comments have been considered with accuracy.

Details comments were given in the attached file.

General comments

All abbreviations (e.g. CVD, CKD, SFA, BP, DM, RCT, etc.) need to be defined at the first mention in the abstract and the main text. Please check the entire manuscript carefully.

Thank you for your recommendation. The manuscript has been thoroughly checked by two members of the team and all abbreviations amended. See SEIFA defined line 644, cardiovascular diseases (CVD) line 223, glomerular filtration rate (GFR) line 244, Statistical Package for Social Sciences (SPSS) line 372, one-way analysis of variance (ANOVA) line 634, food frequency questionnaire (FFQ) line 1150

In the aim of the study, you use 'markers of non-communicable diseases' without giving any markers.

Please note that 'cholesterol status' is not a disease. Did you mean dyslipidemia? It is also a marker of cardiovascular diseases. Please reformulate the aim of the work. Please note that the same aim of the study should be in both the abstract and the main text of the manuscript.

Thank you for this suggestion. There is now a consistent aim in the abstract (lines 16-19) and at the end of the introduction (lines 253-257). We refer to cardiovascular diseases, not “cholesterol status”.  We have removed the term “cholesterol status’ throughout the document and in tables 3 and 5 to avoid confusion.

Abstract

Line 20-21: I propose to provide: 'were in the middle tertile of the MD score' instead of: 'were in the middle tertile for MD adherence scores'

This has been amended in line 20.

Line 23-24: Please precise describes the major results (provide values) regarding the associations between MD adherence and the risk of non-communicable diseases and their markers.

This has been adjusted in lines 23-24.

Line 25-27: The conclusions are too general, it should be more closely related to the results obtained in the study. Conclusions can not be generalized to 'non-communicable diseases' because not all possible diseases were examined in this study. The used word 'additionally' does not match the conclusions.

Thank you for your suggestion. The conclusions have been significantly reworked and a new paragraph has been added, see lines 26-29

Introduction

Line 80: a paragraph indentation is needed.

Done

In the explain the proposed associations between MD and the risk of chronic diseases and the elevated levels of their markers, you need to give the readers an understanding of the types of studies that have been done and how reliable this evidence is. Please leave the level of evidence and transfer the details of possible mechanisms of these associations into the 'Discussion' section.

Please note that these have now been moved to the discussion- lines 1135-1197

Materials and Methods

After reading section 2.1. 'Study population' I suggest changing its title into the 'Study design and sample collection'.

Amended accordingly . line 260

Section no. 2.2. 'Biochemical data' needs to be completed with necessary methodological details. Please provide all examined biomarkers, methods of their measurements and levels for diagnosis of non-communicable diseases. This is very important information and quoting is not enough. What about CKD biomarkers?

Kindly note that this section has now been modified, lines 383-401

CKD biomarkers in lines 398-400

Please describe the ‘dyslipidemia status’ in the 'Biochemical data' section.

Thank you for this note, we have added this to the 'Biochemical data' section. Please see lines 392-395.

There was no about BMR calculate or measure (see Table 1 in the Results section) - please provide methodological details in the Methods section.

BMR was predicted by their age, sex and weight. This is now in the methods section line 397-398.

Line 383-385: Does an ethanol source, e.g. wine, drinks, etc. matter in this context?

In Trichopoulou’s score, the source of alcohol does not matter – we have added a note about this in line 622.

In the 'Statistical analysis' section, please provide the kind of statistical test used to assess ORs in the logistic regression analysis.

We used a multi-variate logistic regression. We used Chi square test for OR. This has now been noted in the text line 638

Results

There are missing:
- MD tertiles (range of score's) (Table 1-5);
- data for the total sample (Table 1-4);

Kindly note that for table 1,2,3,4 we added in a descriptives column for our total sample.

Please consider combine table 2, 3 and 4 in one table. What did you think of this?

Table 1: Why did you enter 'tertile' (in a row) with continuous variables like BMI, waist circumference, percentage of energy from ... etc.? This is unnecessary, just enough 'mean, SD'.

We wanted to see how the mean,sd are differing across different tertiles, hence we reported mean,sd for each tertile alone

Table 1: You can show only one category of waist circumference, which indicate the increased risk of … (please provide values).

Done

Table 2: Please avoid repetition, e.g. enter first the line (header) with the type of disease, and then in the following lines: markers and their levels. This will increase the readability of the table.

Kindly note that table 2 has been amended accordingly.

Table 2: I don't understand why you give HbA1c >= 6.5% twice, first as 'has diabetes mellitus' and then as 'indicated diabetes mellitus' with other values.

Thank you for this suggestion, please note that we now have 1 indicator for HbA1c in table 2

Table 2: I don't see the sample percentage for the elevated CKD markers, which were mentioned in the text.

Kindly note that the % of elevated CKD are present in table 2 for each tertile. We have also added the numbers for the total sample

Table 3: Please provide data only for the elevated/abnormal levels of the lipid profile.

We decided to put on all the results, even the none significant ones, to avoid any kind of reporting bias that may result from only reporting significant results

If the results are statistically significant, compare the values in the description of the results.

Table 5: I don’t see the described results of univariate analysis. You can show results for two models: no adjusted and adjusted and the results for MD tertiles (upper vs. bottom as reference and middle vs. bottom as reference) and for elevated/abnormal level vs. normal level (as reference) of markers.

Apologies, we only conducted a bivariate analysis using Chi Square tests and a multivariate analysis and the wrong term (univariate) was written in line 854 and line 858. We have corrected this and changed univariate to bivariate (referring to the earlier mentioned Chi Square tests). We then present information on the multivariate analysis to adjust for confounders and examine the association between the variables.

Discussion

In the first paragraph, the most important results should be highlighted, especially those statistically significant, but without comparison with literature data.

Thank you for this suggestion – the most important results are highlighted in lines 866-878 at the beginning of the discussion.

Please provide more comparisons of your own results (I mean mainly OR values, but without unnecessary details such as 95% CI) with the results of other studies.

Thank you – this has been modified accordingly – please see additions throughout the discussion section relating to our results and the other studies.

Paragraphs should be divided thematically, e.g. the MD and lipid profile association and the MD and blood pressure association should not be discussed in one paragraph.

Kindly note that this has now been modified and the paragraph has been split to two. Now each paragraph is one theme (MD and one diseases).

The level of evidence described (line: 568-593) should be transferred to the Introduction.

Done – Please see lines 227-246

Please combine the last paragraph of the discussion (line: 635-641) with 'Limitations' into one section 'Study strengths and limitations'.

These have now been combined lines 1400-1406

Conclusions

In the first and second paragraphs, most sentences have the character of Introduction or Discussion. Please formulate conclusions based on the results obtained.

Thank you for noting this, the conclusion section has now been modified- lines 1427-1436